# Promoting Media Literacy Online: An Intervention on Performance and Appearance Enhancement Substances with Sport High School Students

**DOI:** 10.3390/ijerph18115596

**Published:** 2021-05-24

**Authors:** Federica Galli, Tommaso Palombi, Luca Mallia, Andrea Chirico, Thomas Zandonai, Fabio Alivernini, Alessandra De Maria, Arnaldo Zelli, Fabio Lucidi

**Affiliations:** 1Department of Movement, Human and Health Sciences, University of Rome “Foro Italico”, 00185 Rome, Italy; luca.mallia@uniroma4.it (L.M.); de.maria.alessandra@gmail.com (A.D.M.); arnaldo.zelli@uniroma4.it (A.Z.); 2Department of Social and Developmental Psychology, “Sapienza” University of Rome, 00185 Rome, Italy; tommaso.palombi@uniroma1.it (T.P.); andrea.chirico@uniroma1.it (A.C.); fabio.alivernini@uniroma1.it (F.A.); fabio.lucidi@uniroma1.it (F.L.); 3Sports Research Centre, Department of Sport Sciences, Miguel Hernández University, Av. de la Universidad s/n, Elche, 03202 Alicante, Spain; thomaszandonai@gmail.com

**Keywords:** media literacy, performance and appearance enhancement substances, doping, sport high school students, COVID-19, online delivery, social media

## Abstract

The outbreak of coronavirus required adjustment regarding the delivery of interventions. Media literacy interventions are necessary to help people acquire relevant skills to navigate the complexities of media communications, and to encourage health-promoting behaviors. The present study aimed to promote a media literacy intervention regarding performance and appearance enhancement substances use in sports high school students. The COVID-19 contingency allowed us to evaluate whether online sessions can effectively promote greater awareness of media influence, a stronger sense of confidence in persuading others to deal with media messages, and healthier attitudes about PAES use among high school students. The study relied on an “intervention group” comprising 162 students (31.5% female) and a “control group” comprising 158 students (42% female). Data were analyzed through repeated measures of Group X Time MANOVA and ANOVA, demonstrating some degree of efficacy of the media literacy intervention. The “intervention group” reported higher awareness of potential newspapers’ influence and a significant increase in their sense of confidence in dealing with media influence compared to the “control group”. Findings support the efficacy of online media literacy programs to prevent doping consumption in adolescents.

## 1. Introduction

In recent years, technological progress and multimedia devices have caused a radical change in society, especially concerning people’s access and use of web-based and internet services and information. Furthermore, the current coronavirus (SARS-CoV-2) worldwide pandemic and the consequent restrictive measures adopted to contain it (e.g., quarantine, lockdowns, social distancing) have heightened people’s exposure to multiple forms of media-based news, messages, and information.

Several studies have amply demonstrated that media messages play a fundamental role in shaping psychological processes and human behavior [1,2,3]. In light of the surge in people’s wide exposure to web-based information that technological advances, the use of online platforms, and even the current pandemic have certainly contributed to, it seems particularly urgent to develop specific educational media literacy interventions that can effectively help people to acquire a set of knowledge and skills for a correct understanding and use of media-based news and information (see metanalysis of [4]).

This type of educational program typically aims at teaching people the best ways to identify, analyze, and manage media messages encouraging health-promoting behaviors [5]. Overall, interventions tend to focus on youth and their abilities to correctly identify the type of consumer and the corresponding messages that advertisements, blogs, or websites may systematically target and promote. Interventions of this sort and their evidence-based efficacy have been the focus of studies addressing a variety of health-related behaviors, ranging from substance use/abuse (e.g., [6,7]), to use of tobacco (e.g., [8]) and risky sexual behaviors (e.g., [9]).

Other studies followed a similar approach and focused on the role media may play in eliciting the use of doping substances (e.g., [10,11]), distinguishing between the use of “controlled” and “uncontrolled” performance and appearance enhancement substances (PAESs). The first class refers to doping substances prohibited in competitive sports, as established by regulations of the World Anti-Doping Agency (WADA), and whose use implies legal sanctions. Existing studies of this type of PAES focused on adolescents (e.g., [12] Johnston et al., 2007) and high-level athletes (e.g., [7]), as well as on non-athletes (e.g., [13]). On the other hand, “uncontrolled” PAESs can be freely purchased without any restrictions and with no legal or social sanctions ensuing from their use (i.e., proteins, creatine), and existing studies of this latter type of PAES have focused on athletes at different competitive levels (i.e., elite, amateur, and recreational) and on different ages [14].

A core theme of the scientific literature addressing media literacy interventions on the use of PAESs is that of media “power” or influence. For instance, a high volume of media messages tends to promote the use of these substances as a positive and effective strategy to achieve personal goals, such as the desire to enhance one’s performance and/or physical appearance [11]. Likewise, media may contribute to creating role models among athletes who have gained popularity and been users of doping substances, thus contributing to a process of social influence toward supporters’ adoption of risky health behaviors or choices to consume PAESs [11,15].

### The Present Study

The present study departed from the above considerations and followed the core protocols adopted by Mallia and colleagues (2020) [11] and their media literacy intervention on PAES use conducted with sports science university students. The present study focused, however, on adolescents and on sport high school students in particular. This focus stemmed from existing evidence that the highest use of PAESs tends to be among adolescents practicing sport [7,13,16] and from the general notion that extensive use of social media by adolescents tends to be associated with high rates of risky health behaviors, including substance consumption [17].

The present study also differed from Mallia and colleagues’ 2020 [11] study in that it attempted to implement a media literacy intervention among sport high school students in the novel context of online web-based teaching. Indeed, the study was conceived during the initial stages of the COVID-19 pandemic and in the context of the protocols that schools had to follow to guarantee teaching and learning for their students. This contingency offered the unique opportunity to evaluate whether media literacy interventions can be effectively delivered online. More specifically, the present study focused on whether teaching sessions that are offered online can be effective in promoting greater awareness of media influence, a stronger sense of confidence in persuading others to deal with media messages, and healthier attitudes about PAES use among high school students.

## 2. Materials and Methods

### 2.1. Participants and Procedures

Eight Italian sports high schools were contacted and fully informed about the aims of the present research. The study was approved by the Ethics Review Board of the Department of Social and Developmental Psychology, “Sapienza” University of Rome. The study relied on written consent for participation (if minors, parents provided consent), and recruitment procedures led to the selection of two distinct groups of students. One-hundred and sixty-two sport high school students (31.5% female; *mean*_age_ = 16.93; *SD* = 0.87; *range*_age_ = 15–18) actively participated in the intervention (i.e., “intervention group”), whereas 158 high school students (42% female, *mean*_age_ = 15.62; *SD* = 1.33; *range*_age_ = 13–19) participated in the study by only contributing to the questionnaire assessment (i.e., “control group”). All students were recruited using a convenience sampling procedure and they were assigned to the two groups (i.e., “intervention” and “control”) based on the schools’ availability (i.e., convenience assignment).

### 2.2. Questionnaire Assessment

All students provided data in two separate sessions on a set of media literacy measures and pro-doping attitudes, that is, both before and after the scheduled intervention sessions involving students in the intervention group. The set of questionnaires included all the media literacy measures previously used in Mallia and colleagues’ study (2020) [11] and targeted the following variables:(1)Three “media influence” four-item sets referring, respectively, to social media, TV, and newspapers, measured the extent to which students were aware of the influence media might have on beliefs and behaviors concerning PAES use in sport [11,18]. A sample set of three items was “*Social Media/TV/Newspaper messages affect the way young people think about the use of substances to enhance their physical appearance*”. Students declared their level of agreement considering a 7-point response scale ranging from 1 (“Totally disagree”) to 7 (“Totally agree”).(2)Four “media realism” four-item sets referring, respectively, to TV, newspapers, websites, and social media measured the extent to which students considered media to be realistic sources of information for youths’ behavior [11,19]. A sample set of four items was “*TV/Newspapers/Websites/Social Media are realistic sources of information for how people my age act*”. Students answered using a 7-point response scale ranging from 1 (“Never”) to 7 (“Always”).(3)The sense of confidence students may personally have in persuading others to effectively deal with media messages (i.e., perceived self-efficacy) was assessed using a 3-item scale [11,19]. A sample item was “*I have ideas about how I can use media to affect whether other teenagers use substances to enhance their performance or physical appearance*.” Students declared their level of agreement using a 7-point response scale ranging from 1 (“Totally disagree”) to 7 (“Totally agree”).(4)Finally, a set of six semantic differential items measured the extent to which students endorsed positive attitudes toward PAES use [10,11,20,21]. Students rated on a five-point scale the extent to which the “*use of PAES would be…*” useless/useful, foolish/wise, undesirable/desirable, negative/positive, harmful/beneficial, and advantageous/disadvantageous.

### 2.3. Intervention and Its Implementation

The intervention was developed, coordinated, supervised, and monitored by researchers from the University of Rome, “Sapienza” and University of Rome, “Foro Italico”. The design of the intervention closely followed the structure that was implemented in the high school study of Lucidi and colleagues (2017) [10] and in the sports science university study of Mallia and colleagues (2020) [11].

Specifically, the intervention protocol included two phases that were carried out in the fall of 2020 during a time in which students participated in their regular school teaching program from home due to the COVID-19 restrictions imposed by national state agencies. In the first one, each of three experts (a sport sciences expert, a communication expert, and a sport psychologist) led two 90 min online sessions addressing the topic of media messages concerning PAES use in sport from a personal disciplinary standpoint.

The second phase consisted of six online workshop-like sessions (90 min per session) in which a trained psychologist encouraged and supervised students during personal work. All sessions were run using web platforms (e.g., Google Meet, Microsoft Teams, Zoom) that students typically used for their daily school activities.

As to the specific content of the first phase’s online class sessions, the sport sciences expert focused on not only the moral and ethical consequences of PAES consumption but also the effects regarding the damage to health. The sports science expert focused on how media may minimize these issues. The communication expert instead explained to participants the role of media messages in fostering biased beliefs about PAES consumption. The communication expert underlined how these biased beliefs promote unrealistic sport and aesthetic goals. Finally, the sport psychologist described the ways cognitive strategies may help students correctly analyze and evaluate media messages, encouraging the awareness of their personal sports goals and the capacity to counteract possible temptations toward the consumption of PAESs. As to the activities encouraged during the second phase’s online sessions, the trained psychologist initially called students’ attention to the biases and types of justifications that media typically boost and reinforce in their messages concerning sport and PAES use, as well as to the best ways to recognize and counteract them. Then, the intervention involved the division and organization of small groups of students to design and implement an “awareness-raising campaign” or a media message against PAES use. These small group activities focused on creating outputs (e.g., text, video, pictures) that could easily be published and shared online through the main social media platforms (e.g., Facebook and Instagram). Following the sport psychologist’s instructions, the students were invited to organize their products at home. The sport psychologist monitored the students’ progress during the subsequent sessions to revise and correct their ideas and/or outputs and answer their doubts and questions.

### 2.4. Data Analysis

First, the reliability (i.e., Cronbach’s alpha) of the media literacy measures was examined. Then, the analyses focused more prominently on the expected effects of the media literacy intervention implemented online. To test the possible intervention effects, two distinct repeated measures, “Group by Time” MANOVA, first evaluated whether intervention group students, as compared to the control group, reported greater average changes in the levels of, respectively, awareness of media influence and realism of media messages across assessments (i.e., before and after the intervention). Similarly, two additional repeated measures, “Group by Time” ANOVA, evaluated whether the intervention yielded possible changes in sport high school students’ personal sense of confidence and attitudes toward PAES use.

## 3. Results

In terms of the reliability, the alpha coefficients of all the measurements support the internal consistency of the item sets (range from 0.64 to 0.91).

### 3.1. The Efficacy of the Media Literacy Intervention

#### 3.1.1. Awareness of Media as Sources of Influence

At a multivariate level, there was no statistically significant “Group by Time” interaction effect, nor a “Time” main effect on students’ awareness of media influence (*Wilks’ Lamba* _(3, 316)_ = 0.993; *p* = 0.506; *partial eta squared* = 0.007; *Wilks’ Lamba* _(3, 316)_ = 0.981; *p* = 0.111; *partial eta squared* = 0.019, respectively). Furthermore, a significant multivariate main effect for “Group” (*Wilks’ Lamba* _(3, 316)_ = 0.976; *p* = 0.05; *partial eta squared* = 0.024) was observed. There were also two statistically significant univariate main “Time” effects for students’ awareness of social media (*F* _(1, 318)_ = 4.51, *p* = 0.035; *partial eta square* = 0.014) and newspaper (*F* _(1, 318)_ = 4.29, *p* = 0.039; *partial eta square* = 0.013) influences.

When the latter two univariate effects were more carefully examined via pairwise comparisons, differences across pretest and posttest scores seemed to be relatively more pronounced among intervention group students. In particular, these students on average showed a significant increase over time in their awareness of newspaper influences (*F* _(1, 318)_ = 3.73, *p* = 0.05; *partial eta square* = 0.012; *Mean* _(time 1)_ = 2.49; *SD* _(time 1)_ = 1.13, *Mean* _(time 2)_ = 2.70; *SD*
_(time 2)_ = 1.26), and a close to significance increase in their awareness of TV influences (*F* _(1, 318)_ = 3.65, *p* = 0.06; *partial eta square* = 0.011; *Mean* _(time 1)_ = 2.75; *SD* _(time 1)_ = 1.25, *Mean* _(time 2)_ = 2.96; *SD* _(time 2)_ = 1.35) and social media influences (*F* _(1, 318)_ = 3.24, *p* = 0.07; *partial eta square* = 0.01; *Mean* _(time 1)_ = 3.47; *SD* _(time 1)_ = 1.41, *Mean* _(time 2)_ = 3.69; *SD* _(time 2)_ = 1.34). There were no significant or close to significance pairwise comparisons indicating possible time effects in media awareness among control group students.

#### 3.1.2. Perceived Realism of Media Information

With respect to students’ perceived level of realism in media, the results again showed no statistically significant “Group by Time” interaction (*Wilks’ Lamba* _(4, 315)_ = 0.991; *p* = 0.608; *partial eta square* = 0.009), no significant multivariate main effect of “Time” (*Wilks’ Lamba* _(4, 315)_ = 0.994; *p* = 0.765; *partial eta square* = 0.006), nor a significant multivariate main effect for “Group” (*Wilks’ Lamba* _(4, 315)_ = 0.985; *p* = 0.295; *partial eta square* = 0.015). There was no statistically significant effect of time across the two experimental groups for pairwise comparisons.

#### 3.1.3. Students’ Self-Efficacy

The results of an ANOVA of students’ perceived sense of confidence in dealing with media messages indicated a statistically significant “Group by Time” interaction effect (*F* _(1, 318)_ = 18.52; *p* < 0.001; *partial eta square* = 0.055). To visualize this interaction effect, Figure 1 shows the students’ self-efficacy mean scores across experimental conditions and across timepoints. As one can also see, intervention group students reported a statistically significant increase in their average level of self-confidence over time (*F* _(1, 318)_ = 49.41; *p* < 0.001; *partial eta square* = 0.134), whereas there was no statistically significant change in perceived self-efficacy over time among control group students (*F* _(1, 318)_ = 0.318; *p* = 0.372; *partial eta square* = 0.003).

#### 3.1.4. Positive Attitudes toward PAES Use

Finally, the results of an ANOVA on students’ levels of positive attitudes toward the use of PAESs yielded no statistically significant effect for the “Group by Time” interaction (*F* _(1, 318)_ = 0.152; *p* = 0.697; *partial eta square* < 0.001) and the main effects of “Time” (*F* _(1, 318)_ = 2.303; *p* = 0.765; *partial eta square* = 0.007) and of “Group” (*F* _(1, 318)_ = 0.117; *p* = 0.733; *partial eta square* < 0.001).

## 4. Discussion

Media literacy stands as an efficient educational strategy to promote youths’ conscious analysis and management of media messages and a critical thinking approach to their potential influence (e.g., [5,22]). With this general notion in mind, the authors of the present research focused their attention on adolescents’ use of “performance and appearance enhancement substances” (PAESs) in sport contexts [7,13], and aimed at replicating the media literacy intervention of Mallia and colleagues (2020) [11].

Concerning the original intervention, the present study introduced two key novelties. The present media literacy intervention was conducted with sport high school students rather than sports science university students. The second novelty was educationally challenging in that the study was conducted during the pandemic of COVID-19, and it required the adaptation of the intervention’s in-person teaching protocols to the restrictions of web-based online teaching imposed by the pandemic. For these reasons, the intervention sessions, which included a series of teaching classes conducted individually by three specialists (i.e., in the order of teaching, they were a sport sciences expert, a sports communication expert, and a sport psychologist) were modified to maximize students’ learning, heavily focused on comprehensible language, visual aids, and examples, and were organized to be easily accessible online (i.e., sessions were implemented on different web platforms).

Based on these premises, the main goal of the present research was to evaluate, for the first time, the efficacy of an “online” media literacy intervention among sports high school students. Empirically, the goal was to assess whether the intervention would yield changes in young students’ awareness of media as sources of influence, their perceived realism of media messages, their personal sense of confidence or self-efficacy in persuading others to deal effectively with media messages and their potential influence, and their attitudes towards PAES use. Following a traditional design, and in line with the study of Mallia et al. [11], these empirical goals were pursued by comparing effects across different forms of media (e.g., newspapers, TV, and social media platforms), by comparing group effects across students who participated in the intervention sessions and students who did not, and by assessing possible averaged changes in scores collected before and after the intervention sessions.

Overall, the findings showed some degree of efficacy of the media literacy intervention. Students who participated in the intervention reported relatively higher awareness of potential newspaper influences and, albeit marginally, of TV and social media influences. This finding might plausibly be due to the nature of the teaching led by the communication specialist who, being a journalist and a particular expert in sports events and doping cases reported by traditional newspapers, may have contributed to students’ heightened awareness of newspapers as one of the main sources of influence on PAES consumption. Speculatively, we presume that the many examples utilized during teaching and which may have glorified athletes who experienced doping use and their “fake sports performances” made a particularly strong impression on young students.

Compared to their counterparts, students who participated in the intervention also experienced a significant increase in their sense of confidence in dealing with media influence. This finding is in line with evidence reported by Jeong and colleagues in their 2012 meta-analysis (*d* = 0.34, *p* < 0.001, 95% *CI*: 0.18 to 0.50) [4]. The finding is also quite consistent with Mallia and colleagues’ findings (2020) [11], showing once again the efficacy of educational media literacy programs in promoting a sense of personal confidence among adolescents and young adults.

There were no detectable intervention effects on positive attitudes towards PAES use. Although this finding was unexpected, it solicits several considerations. In retrospect, it is quite plausible that the proposed intervention focused too heavily on the issues of media influence and fake news and did not carefully balance the core intervention themes of media influence and PAES use. This clearly highlights the importance of upgrading our media literacy intervention to render it educationally more effective. It is also possible or plausible that implementing an “online” media literacy intervention program, rather than “in-person” teaching, substantially limited the chances of changing sport high school students’ attitudes about the personally relevant and sensitive issue of PAES use.

### Strengths, Limitations, and Future Research

The present research has several strengths. First, this study represents the first online media literacy program implemented with sports high school students, a group which is, because of age and specific context, particularly at risk for PAES use (e.g., [7]). Secondly, the study considered a sample of students nationwide, covering regions and schools distributed in the north, center, and south of the country. Third, the media literacy intervention was implemented online by using different web platforms (i.e., Google Meet, Zoom, and Microsoft Teams). This latter strength is particularly relevant if we consider that the implementation of online intervention could be a strategy to break down environmental, physical, and social barriers.

It is essential to note the different limitations of the present research. First, the investigation was implemented in high school settings, and, as a result, the two samples of sports high school students were not stratified samples. Therefore, findings should not and cannot be generalized to the broader sports high school student population. Secondly, students’ assignment to the two conditions of the media intervention program was not rigorously randomized, but was based on the schools’ availability by using a convenience sampling procedure. Finally, if it is true that the online delivery could be considered as an important strength of the present research, on the other side, this modality may present some limitations, for example, in terms of technological limits (e.g., a lack of Internet connectivity and electronic devices), pedagogical disadvantages (e.g., the difficulty to stimulate proactive behaviors), or social restrictions (e.g., the loss of human interaction among students [23]).

Future studies should implement media literacy interventions in sport high schools in a face-to-face intervention, with the scope of comparing the two modalities (i.e., online vs. face-to-face). Future research should also consider participants’ emotions, cognitions, and behaviors elicited by their exposure to media influence, emphasizing PAES use [24,25].

## 5. Conclusions

As a final consideration, these latter remarks seem particularly important if one considers that the existing literature shows that PAES consumption is particularly frequent among adolescents across all levels of sport and physical activity [7,10]. Sport high school students represent, therefore, a sensitive risk group, and media literacy programs might stand as an important tool to prevent PAES use elicited, perhaps even unintentionally, by the intricacies of media messages and by social media’s powerful influence.

## Figures and Tables

**Figure 1 ijerph-18-05596-f001:**
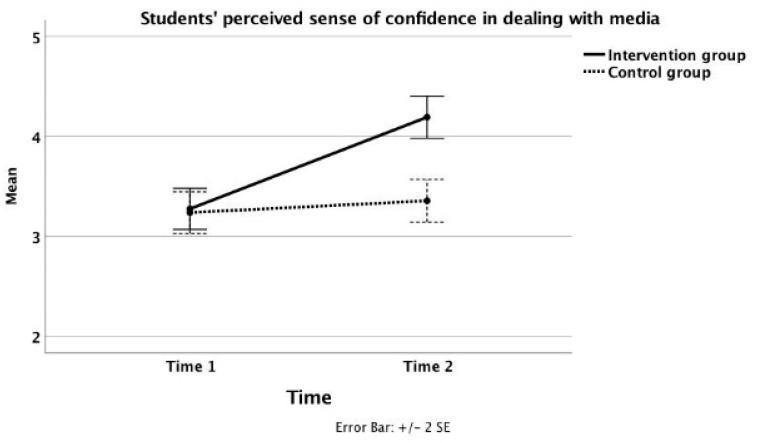
The students’ self-efficacy across experimental conditions and across timepoints.

## Data Availability

The data presented in this study are available on request from the corresponding author.

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
