# Peer review of "Promoting Media Literacy Online: An Intervention on Performance and Appearance Enhancement Substances with Sport High School Students"

_ijerph, 2021, doi:10.3390/ijerph18115596_

Round 1

Reviewer 1 Report

General comment

The authors present an interesting and valuable study in which they evaluated the effectiveness of an online media-literacy intervention regarding the use of Performance and Appearance Enhancement Substances in sport high school students. The results highlighted that, compared to the control group, the students who took part in the intervention reported higher awareness of potential newspapers influence and a significant increase in their sense of confidence in dealing with media influence. These outcomes suggest the importance to promote media-literacy programs to prevent the doping consumption in adolescents, delivered also in online modalities.

The manuscript is well-written, the methodology is appropriate, and the outcomes significantly add to the literature concerning this line of research.

My evaluation of this manuscript is positive and I have just some minor suggestions that in my opinion will help authors to further improve the overall quality and clarity of their work.

Detailed comments

P2, L99: it would be useful if authors could provide details about how the “selection of two distinct groups of students” was performed.

P3, L115 & 128: please replace “our” with “their”.

P3, L148: it is clear that the intervention protocol was followed exclusively by the intervention group, thus writing ‘“intervention group” students …’ only for the second phase (i.e., after not doing so for the first one) may cause some confusion. To avoid this potential ambiguity, I suggest the authors to rephrase the description of the second phase, e.g., “The second phase consisted of six online laboratory-like sessions…” without mentioning the group label.

In the same line: maybe “workshop-like” is more appropriate than “laboratory-like”?

Moreover, authors could provide some details: a) the duration of the laboratory-like sessions, and b) whether (and how) students’ attendance to the various activities was monitored, maybe reporting this descriptive datum (if they have it, of course).

P4, L183: Is the word “key” (referred to measurements) necessary? I mean, if there wasn’t a distinction between key and non-key measurements, maybe it is not necessary.

P4, L192-193 vs 200-201: there are two occurrences of p = .05: the former is considered as close to significance, while the latter as significant; please be consistent.

P5, L239-242: please remove this part.

P6, L293: when I read “fake news”, the gamified inoculation approach (e.g., Basol et al., 2020) came to my mind; authors may consider such an approach (or similar ones) for their future interventions.

P6, L301: as this subsection starts with the strengths, I suggest to rename it “Strengths, limitations, and future research” (or future perspectives).

P7, L306-308: also in light of what is stated on L315-319, authors should better specify why the fact that the intervention was implemented online should be considered as a strength (I agree with it, but authors could be clearer).

References

Basol, M., Roozenbeek, J., & van der Linden, S. (2020). Good news about bad news: Gamified inoculation boosts confidence and cognitive immunity against fake news. Journal of cognition, 3(1). https://doi.org/10.5334/joc.91

Author Response

Dear Reviewer,

Thank you for your general comments and your accurate suggestions. They represent an important starting point to improve our paper.

You can find our answer in bold type in referring to each point:

P2, L99: it would be useful if authors could provide details about how the “selection of two distinct groups of students” was performed.

Thank you for your comment. In light of this, we added a sentence to explain the convenience sampling procedure: “All students were contacted using a convenience sampling procedure, and they voluntarily decided to participate in this research”. 

P3, L115 & 128: please replace “our” with “their”.

We provided to replace based on your comment. 

P3, L148: it is clear that the intervention protocol was followed exclusively by the intervention group, thus writing ‘“intervention group” students …’ only for the second phase (i.e., after not doing so for the first one) may cause some confusion. To avoid this potential ambiguity, I suggest the authors to rephrase the description of the second phase, e.g., “The second phase consisted of six online laboratory-like sessions…” without mentioning the group label.

Thank you for your comments. To avoid potential ambiguity, we followed your suggestion to rephrase the description of the second phase.

In the same line: maybe “workshop-like” is more appropriate than “laboratory-like”?

We agree with your suggestion. So, we decided to use "workshop-like" instead of "laboratory-like".

Moreover, authors could provide some details: a) the duration of the laboratory-like sessions, and b) whether (and how) students’ attendance to the various activities was monitored, maybe reporting this descriptive datum (if they have it, of course).

Thank you for your comments. To clarify the details regarding the laboratory-like sessions, we specified the duration of these sessions, adding "(90 min per session)". To describe the monitoring process, we added the present paragraph: "Following the sport psychologist’s instructions, the students were invited to organize their products at home. The sport psychologist monitored the students’ advances during the subsequent sessions to revise and correct their ideas and/or outputs and answer their doubts and questions”. 

P4, L183: Is the word “key” (referred to measurements) necessary? I mean, if there wasn’t a distinction between key and non-key measurements, maybe it is not necessary.

We agree with your suggestion. Therefore, we decided to delete the distinction between key and non-key measurements. 

P4, L192-193 vs 200-201: there are two occurrences of p = .05: the former is considered as close to significance, while the latter as significant; please be consistent.

Thank you for your comment. We rephrased the text to be consistent and considering that p = .05 refers to significant results.

P5, L239-242: please remove this part.

Thanks to your suggestion, we delete the lines from 239 to 242. 

P6, L293: when I read “fake news”, the gamified inoculation approach (e.g., Basol et al., 2020) came to my mind; authors may consider such an approach (or similar ones) for their future interventions.

Thank you for your suggestion. The Basol and colleagues' study is very interesting. It could be a valid starting point for research that focuses on fake news in general and uses gaming intervention. Therefore, for this reason, we would like not to insert the suggested reference to avoid taking into consideration interventions that probably differ too much from our type of intervention.

P6, L301: as this subsection starts with the strengths, I suggest to rename it “Strengths, limitations, and future research” (or future perspectives).

In line with your suggestion, we rename the paragraph “Strengths, limitations, and future research”.

P7, L306-308: also in light of what is stated on L315-319, authors should better specify why the fact that the intervention was implemented online should be considered as a strength (I agree with it, but authors could be clearer).

Thank you for your comment. Considering your indication, we specified, “This latter strength is particularly relevant if we consider that the implementation of online intervention could be a strategy to break down the environmental, physical and social barriers”. 

References

Basol, M., Roozenbeek, J., & van der Linden, S. (2020). Good news about bad news: Gamified inoculation boosts confidence and cognitive immunity against fake news. Journal of cognition, 3(1). https://doi.org/10.5334/joc.91

Reviewer 2 Report

Thank you for the opportunity to review your paper reporting the findings from your study of the effects of an online media-literacy intervention to promote awareness of media influence, confidence in persuading others to deal with media messages, and healthier attitudes about PAES.   

Written expression is problematic throughout. I have detailed here some instances where corrections are needed just the abstract alone. There are many other instances in the main body of the manuscript requiring correction. I have attached a marked-up copy of the manuscript indicating where various (but not all) corrections are required. 

Abstract:

l 12. “The outbreak of the Coronavirus led people to be particularly exposed to internet information.” Exposure to internet information has been prevalent for the lifetimes of anyone born in the current century, as is the case for the high-school-aged students in the current study. Rather than internet exposure coming as a consequence of the pandemic, the manner of performing research and interventions required some rapid adjustment.

ll 12-13. “Then, comes the need to…” Grammar check

ll 13-14 “…to help people acquire skills for a correct use of media”. What is a ‘correct use of media’? Correct assumes a dichotomy of correct/incorrect, whereas media use spreads across a spectrum of appropriateness depending on context and circumstances.

l 16. “Covid-19”. COVID-19 according to the WHO, who named the disease https://www.who.int/emergencies/diseases/novel-coronavirus-2019/technical-guidance/naming-the-coronavirus-disease-(covid-2019)-and-the-virus-that-causes-it

ll 18-19. “The study relied on an “intervention group” composed by 162 students (31.5% female) and a “control group” comprised of 158 students (42% female).” REPLACE. Composed by and comprised of both incorrect.

l 20. “Data were analyzed testing repeated measure “Group by Time” MANOVAs and ANOVAs…” Should be “repeated measures Group x Time MANOVA and ANOVA.” The acronyms should not be pluralized. The plural form of analysis of variance is also analysis of variance.

l 23-24. “Findings suggest the importance to promote media-literacy programs to prevent the doping consumption in adolescents, delivered also in online modalities.” Sentence requires rewording for correction and clarity of expression.

In addition to matters of written expression, I have some questions about the study as follows:

Materials and Methods

  1. 3. How was assignment to intervention and control groups managed? Was there random assignment? What was the age range of participants? From means/sd, appears to be from 14-18 years? This is an important detail given that you’re claiming this as a novel feature of your study (p. 6. l 250-251) and also because you indicate that these are “very young students” (p. 6, l 282). I wouldn’t categorise 14-18 year olds as ‘very young’.

I wonder about the control condition here, with students in that group not being engaged in way other than to complete the questionnaire twice. Under those conditions, it seems that we must consider that any reported effects come as the result of engagement versus non-engagement. It seems it would have been better to manage them as a wait-list control group and to have them engage in the intervention afterwards and testing them again, to see if the intervention also worked on them.

Results

  1. 4. Why 'close to significance' main effect for Group (ll 192-193) and yet 'significant' for awareness of newspaper influence (ll 200-201)?

Statistical notations should be italicized.

Discussion

  1. 6. “In retrospect, it is quite plausible that the proposed intervention focused too heavily on the issues of media influence and fake news and did not carefully balance the core intervention themes of media influence and PAES use.”

Given the timing and the frequent advice concerning fake news with respect to COVID-19 spread and suitable precautionary measures, it is possible that the focus could have been taken away from PAES use due to more directly-concrete concerns about personal health and well-being.

Limitations and Future Research

  1. 7. “Secondly, students’ assignment to the two conditions of the media intervention program was not rigorously randomized.”

This is the first mention of randomization, which relates to my previous query as to how assignment to the different groups was managed. This needs to be explained in the method section.

Author Response

Dear Reviewer,

Thank you for your general comments and your accurate suggestions. They represent an essential starting point to improve our paper.

Following your indication in the attached marked-up copy, we generally check grammar and expression along with the manuscript, particularly in the abstract section.

You can find our answer in bold type in referring to each point:

Abstract:

l 12. “The outbreak of the Coronavirus led people to be particularly exposed to internet information.” Exposure to internet information has been prevalent for the lifetimes of anyone born in the current century, as is the case for the high-school-aged students in the current study. Rather than internet exposure coming as a consequence of the pandemic, the manner of performing research and interventions required some rapid adjustment.

Thank you for your suggestion. Your point of view is extremely interesting and, therefore, we considered rephrasing the sentence: "The outbreak of the Coronavirus required rapid adjustment regarding the delivery of interventions".

ll 12-13. “Then, comes the need to…” Grammar check

Considering your suggestion, we checked the grammar.

ll 13-14 “…to help people acquire skills for a correct use of media”. What is a ‘correct use of media’? Correct assumes a dichotomy of correct/incorrect, whereas media use spreads across a spectrum of appropriateness depending on context and circumstances.

Thank you for your comments. Your consideration led us to reflect on this topic and to rephrase the sentence as followed:

“There is the need to develop media-literacy interventions to help people to acquire skills for the intended use of media, encouraging health-promoting behaviors”.

l 16. “Covid-19”. COVID-19 according to the WHO, who named the disease https://www.who.int/emergencies/diseases/novel-coronavirus-2019/technical-guidance/naming-the-coronavirus-disease-(covid-2019)-and-the-virus-that-causes-it

Following your suggestion, we were also consistent in the Abstract section to the WHO's guidelines.

ll 18-19. “The study relied on an “intervention group” composed by 162 students (31.5% female) and a “control group” comprised of 158 students (42% female).” REPLACE. Composed by and comprised of both incorrect.

Considering your accurate general comments regarding the grammar check, we also replace these lines as you suggested.

l 20. “Data were analyzed testing repeated measure “Group by Time” MANOVAs and ANOVAs…” Should be “repeated measures Group x Time MANOVA and ANOVA.” The acronyms should not be pluralized. The plural form of analysis of variance is also analysis of variance.

We corrected the acronyms in the Abstract section and along with the manuscript, thanks to your comment.

l 23-24. “Findings suggest the importance to promote media-literacy programs to prevent the doping consumption in adolescents, delivered also in online modalities.” Sentence requires rewording for correction and clarity of expression.

Thank you for your suggestion. We replaced the sentence to be more precise: "Findings support the efficacy of online media-literacy programs to prevent the doping consumption in adolescents."

In addition to matters of written expression, I have some questions about the study as follows:

Materials and Methods

  1. How was assignment to intervention and control groups managed? Was there random assignment? What was the age range of participants? From means/sd, appears to be from 14-18 years? This is an important detail given that you’re claiming this as a novel feature of your study (p. 6. l 250-251) and also because you indicate that these are “very young students” (p. 6, l 282). I wouldn’t categorise 14-18 year olds as ‘very young’.

Thank you for your comments.

Regarding the assignment to intervention and control group, we specified that the procedure followed a convenience sampling procedure; therefore, we added “All students were contacted using a convenience sampling procedure, and they voluntarily decided to participate in this research" sentence in paragraph 2.1.

Regarding the range's age, we added this information, reporting the mean and the standard deviation and the range for each group. Concerning the comments about the participants' age, we agreed to not considering our samples "very young students".

I wonder about the control condition here, with students in that group not being engaged in way other than to complete the questionnaire twice. Under those conditions, it seems that we must consider that any reported effects come as the result of engagement versus non-engagement. It seems it would have been better to manage them as a wait-list control group and to have them engage in the intervention afterwards and testing them again, to see if the intervention also worked on them.

Thank you for your observation; we completely agree with your comment. Please, consider that you had not time and schools' availability to organize the intervention group and the control group based on the waiting list procedure because the COVID-19 pandemic led to less availability from schools involved in the present research. Your comment is a relevant starting point to consider for future studies.

Results

  1. Why 'close to significance' main effect for Group (ll 192-193) and yet 'significant' for awareness of newspaper influence (ll 200-201)?

Thank you for your comment. We rephrased the text to be consistent and considering that p = .05 refers to significant results.

Statistical notations should be italicized.

Following your recommendation, we replace all statistical notations using Italic.

Discussion

  1. “In retrospect, it is quite plausible that the proposed intervention focused too heavily on the issues of media influence and fake news and did not carefully balance the core intervention themes of media influence and PAES use.”

Given the timing and the frequent advice concerning fake news with respect to COVID-19 spread and suitable precautionary measures, it is possible that the focus could have been taken away from PAES use due to more directly-concrete concerns about personal health and well-being. MALLIA

Thank you for your observation. Your idea could be considered a probable explanation, but our study based on previous studies focusing not exclusively on the prevention to influence from social media but also on the use and consumption of PAES from young people.

Limitations and Future Research

  1. “Secondly, students’ assignment to the two conditions of the media intervention program was not rigorously randomized.”

This is the first mention of randomization, which relates to my previous query as to how assignment to the different groups was managed. This needs to be explained in the method section.

Thank you for your comment. According to it, we described the convenience sample procedure in the method section.

Round 2

Reviewer 2 Report

Thanks for your revised manuscript. I've made additional comments/suggestions in the attached file. In particular, the matter of how assignment to conditions was managed has not yet been addressed, and needs to be. 

Author Response

Dear Reviewer,
Thank you again for your comments and suggestions. Following your indication in the attached marked-up copy, we replaced and reviewed the manuscript regarding grammar check and clarity of some sentences.
You can find our answer in bold type in referring to each point:
Abstract section: we followed your accurate suggestions. 
P.2.1
line 96: we added the parentheses, as you suggested;
lines 101-102: we described the recruitment procedure and the selection procedure regarding the two groups, "All students were recruited using a convenience sampling procedure, and they were designated in the two groups (i.e., "intervention" and "control") based on the schools' availability (i.e., convenience assignment)".
P.2.4
We replaced "was conducted" with "are conducted", as you suggested.
P. 3.1
Thank you for your accurate suggestions. We replaced all statistical notation in italics and standardized spaces. 
P.4.1
line 316: we replaced the sentence based on your previous comments regarding the selection procedure (see lines 101-102), "Secondly, students' assignment to the two conditions of the media intervention program was not rigorously randomized, but it is based on the schools' availability by using a convenience sampling procedure.".